# Coordination-induced O-H/N-H bond weakening by a redox non-innocent, aluminum-containing radical

Soumen Sinhababu [1,3], Roushan Prakash Singh [1,3], Maxim R. Radzhabov[1], Jugal Kumawat [2], Daniel H. Ess[2] & Neal P. Mankad [1]

Several renewable energy schemes aim to use the chemical bonds in abundant molecules like water and ammonia as energy reservoirs. Because the O-H and N-H bonds are quite strong (>100 kcal/mol), it is necessary to identify substances that dramatically weaken these bonds to facilitate proton-coupled electron transfer processes required for energy conversion. Usually this is accomplished through coordination-induced bond weakening by redox-active metals. However, coordination-induced bond weakening is difficult with earth's most abundant metal, aluminum, because of its redox inertness under mild conditions. Here, we report a system that uses aluminum with a redox non-innocent ligand to achieve significant levels of coordination-induced bond weakening of O-H and N-H bonds. The multisite proton-coupled electron transfer manifold described here points to redox non-innocent ligands as a design element to open coordination-induced bond weakening chemistry to more elements in the periodic table.

Proton-coupled electron transfer (PCET) reactions of small molecules are central to prospective energy conversion and storage schemes that promise to replace carbon-based fuel sources[1,2]. For example, water splitting to make oxygen and hydrogen requires orchestrated removal of 2H[+] and 2e[−] from the $H_2O$ molecule despite the large bond dissociation free energy (BDFE) of its O-H bonds (113.0 kcal/mol)[3]. In the context of fuel cell technologies, the optimal catalysts for this anodic water oxidation are $IrO_x$ nanomaterials[4] that rely on a precious metal, iridium, with insufficient earth abundance to support the global-scale energy economy[5]. Similarly, prospective use of ammonia as a clean energy source requires three PCET events per $NH_3$ molecule despite its large N-H BDFE (100.3 kcal/mol)[6,7]. PCET reactions also have relevance to frontier areas of organic synthesis[8–10]. Ideally, catalysts could facilitate PCET by weakening X-H bonds (X = OH or $NH_2$) through chemical interactions with the small molecules. However, coordination of $H_2O$ or $NH_3$ to most metal ions induces acidification but not bond weakening, i.e., H[+] transfer but not accompanying e[−] transfer needed for energy transduction. For example, the classic Werner complex, $[Co(NH_3)_6]^{3+}$, has a p$K_a$ of 13 that is significantly

lower than that of ammonia[11], yet it maintains a high $BDFE_{N-H}$ of 105 kcal/mol[12]. Instances of coordination-induced bond weakening[13] (CIBW) often involve acidification of the X-H bond by coordination to a highly reduced metal center, e.g. Ti[III], Mo[I], Bi[II][12,14–16], thus enabling H[+] transfer from the acidified X-H ligand coupled to e[−] transfer from the reducing metal ion during net PCET (Fig. 1a)[3]. In an extreme case, the $BDFE_{O-H}$ of $[Sm(OH_2)_n]^{2+}$ has been estimated to be in the 26-39 kcal/mol range[17,18], and similar behavior is observed upon coordination of $NH_3$ to Sm[II][19].

Given the importance of PCET to renewable energy, it is critical to consider how CIBW can be implemented using the most earth-abundant metals, which in some cases do not readily access low-valent states. In this regard, it is noteworthy that water oxidation during photosynthesis does not follow the CIBW paradigm described above but rather follows a multisite PCET paradigm. Here, $H_2O$ coordinates to a redox-innocent $Ca^{2+}$ ion that is itself incorporated into a redox-active manganese-oxo cluster that collectively forms the oxygen-evolving complex (OEC) of photosystem-II[20,21]. As such, PCET at the OEC involves acidification of $H_2O$ by the $Ca^{2+}$ Lewis acid combined with e[−]

[1]Department of Chemistry, University of Illinois Chicago, Chicago, IL 60607, USA. [2]Department of Chemistry and Biochemistry, Brigham Young University, Provo 84604 UT, USA. [3]These authors contributed equally: Soumen Sinhababu, Roushan Prakash Singh. ✉e-mail: npm@uic.edu

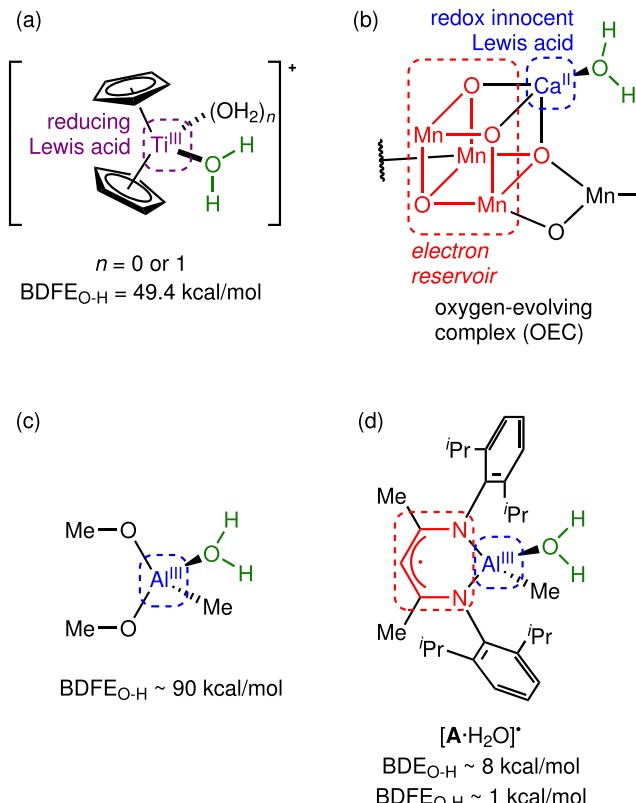

**Fig. 1 | Coordination-induced bond weakening (CIBW) motifs. a** Traditional motif involving a reducing and Lewis acidic metal ion, (**b**) the multisite PCET motif evident in the OEC of photosystem-II, (**c**) lack of weakening with typical $Al^{III}$ ions, (**d**) the multisite PCET scheme reported here relying on redox non-innocence of the ligand. BD(F)E$_{O-H}$ values for Al-OH$_2$ complexes were estimated using DFT calculations.

transfer from the [Mn$_4$] electron reservoir (Fig. 1b), rather than co-localizing H$^+$ and e$^-$ transfer functions at a single metal center.

The most abundant metal on earth is aluminum. Although the common $Al^{3+}$ ion acidifies water dramatically (p$K_a$ = 5.0), typical $Al^{III}$ compounds maintain high BDFE$_{O-H}$ values that are typically >90 kcal/mol. (Fig. 1c) Our group recently discovered that the heterobinuclear complex, LAl(Me)Fp (**1**, L$^-$ = [HC(CMENdipp)$_2$]$^-$, dipp = 2,6-di-*iso*-propylphenyl, Fp$^-$ = [FeCp(CO)$_2$]$^-$), serves as a masked source of the [LAlMe]$^•$ radical (**A**$^•$) that is formally an $Al^{II}$ complex but is better formulated as $Al^{III}$ coordinated by the radical dianion, [L$^•$]$^{2-22,23}$. Since conjugation of the $Al^{III}$ Lewis acid to the [L$^•$]$^{2-}$ electron reservoir bears similarities to the OEC motif (Fig. 1d), we hypothesized that **A**$^•$ would promote CIBW by an analogous, multisite PCET mechanism. Here, we demonstrate CIBW during O-H activation of water/alcohols and N-H activation of an amine at ambient conditions. Not only does this represent a rare example of CIBW by the most abundant metal on earth[24-26], but also computational modeling suggests an unusual degree of CIBW. Most importantly, this discovery points to redox non-innocence[27-29] as a useful design element for enabling PCET with earth's most abundant metals like aluminum and calcium.

## Results

Rapid reactions between **1** and O-H substrates (i.e., H$_2$O and alcohols) or N-H substrates (i.e., NH$_3$ and amines) were observed at ambient conditions (*vide infra*). Thus, the question of reaction pathway immediately arose. Our previous works provided experimental and computational evidence that **1** dissociates reversibly at ambient conditions by Al-Fe homolysis, producing small equilibrium concentrations of the **A**$^•$/Fp$^•$ frustrated radical pair (FRP)[30-32] that can

cooperatively activate *O*-coordinating substrates[22,23]. The majority of available mechanistic data probed CIBW of the C=O π-bonds in CO$_2$ and O=CPh$_2$, the latter of which allowed for spectroscopic characterization of the [LAl(Me)(OCPh$_2$)]$^•$ radical due to stabilizing delocalization of the unpaired spin into the benzophenone π-system[22]. We can now report that this behavior extends to nitrogen-containing π-systems, as well, providing evidence that the FRP is capable of engaging substrates with *N*-coordinating groups. Addition of pyridine to **1** produces Fp$_2$ along with C-C coupled dialuminum complex **2** (Fig. 2a), which presumably forms via pyridine adduct **B**$^•$ that places significant unpaired spin density in the pyridine π-system and triggers diradical coupling at the 4-position. The X-ray crystal structure of **2** indicates localized π-bonding consistent with disruption of pyridine aromaticity: the C2-C3 and C5-C6 distances show double-bond character [1.336(3) Å] while the C3-C4 and C4-C5 distances show single-bond character [1.509(3)–1.516(4) Å], and the N-C2 and N-C6 distances are elongated [1.395(3)–1.400(3) Å] compared to pyridine (1.340 Å)[33]. The C4-C4' distance is also indicative of single bonding [1.572(4) Å], consistent with the pyramidalized, C($sp^3$)-like geometries at these centers.

Having established that the putative FRP is reactive towards both *O*- and *N*-coordinating substrates, the next question was whether observed CIBW of π-systems extends to σ-frameworks. Since we previously showed that **1** is capable of cooperative C-O σ-bond cleavage of cyclohexene oxide[22] and tetrahydrofuran[23], it was implied that intermediate **A**$^•$ induces CIBW of σ-bonds as well as π-bonds. However, far less definitive mechanistic data had been gathered in support of the FRP mechanism for these C-O σ-bond activation reactions involving substrates that lack π-systems to stabilize unpaired spin density. Thus, since σ-bond activation would also be relevant to targeted X-H cleavage by PCET, we sought to establish the FRP mechanism for C-O σ-bond substrate activation by **1**. In particular, we sought to rule out an alternative, polar pathway involving dissociation of **1** by Al-Fe heterolysis to generate the [LAlMe]$^+$/Fp$^-$ frustrated Lewis pair (FLP)[32]. Particularly useful in this regard is the sister compound of **1**, LAl(Me)Wp (**3**, Wp$^-$ = [CpW(CO)$_3$]$^-$), which is reported here. Unlike **1**, which features a direct Al-Fe bond, complex **3** lacks any direct Al-W interaction in favor of an isocarbonyl bridge (i.e. Al⋯O≡C-W, see Fig. 2b). We preliminarily view the structure of **3** as featuring the [LAlMe]$^+$ (**A**$^+$) cation having formed a dative adduct with the Wp$^-$ anion through one of the CO oxygen atoms. In other words, whereas we expected **1** to act as a masked source of the **A**$^•$/Fp$^•$ FRP, we hypothesized that **3** would serve as a masked source of the **A**$^+$/Wp$^-$ FLP in solution.

Accordingly, **1** and **3** were found to exhibit opposite regioselectivity during ring-opening C-O σ-bond activation of (±)-propylene oxide. The reaction of **3** and (±)-propylene oxide produced **4**, which putatively forms via transition state **C** resembling S$_N$2 attack of Wp$^-$ on epoxide coordinated to **A**$^+$ (Fig. 2b). Here, the regioselectivity is consistent with well-known polar epoxide ring-opening pathways that are usually controlled by sterics and, thus, occur at the less substituted carbon center[34]. On the other hand, the reaction of **1** and (±)-propylene oxide produced **5**, which putatively forms via **D** that is the result of coordination-induced C-O σ-bond cleavage by intermediate **A**$^•$ (Fig. 2c). Here, the regioselectivity is dictated by the preference for placing radical character on a secondary rather than primary carbon center in **D**[35]. Collectively, these results are consistent with **1** following an FRP mechanism and **3** following an FLP mechanism during σ-bond activation reactions. Structures of **4** and **5** were assigned definitively by $^1$H NMR spectroscopy, and the assignment of **5** was verified by X-ray crystallography.

Having established that **1** can activate σ-bonds through FRP pathways, next we closely examined the reactions between **1** and substrates containing X-H bonds (Fig. 3a). At temperatures ranging from −30 °C to room temperature, rapid reactions were observed between **1** and the O-H substrates H$_2$O, MeOH, $^i$PrOH, and $^t$BuOH as well as the N-H substrate $^i$BuNH$_2$. Monitoring these reactions by in situ

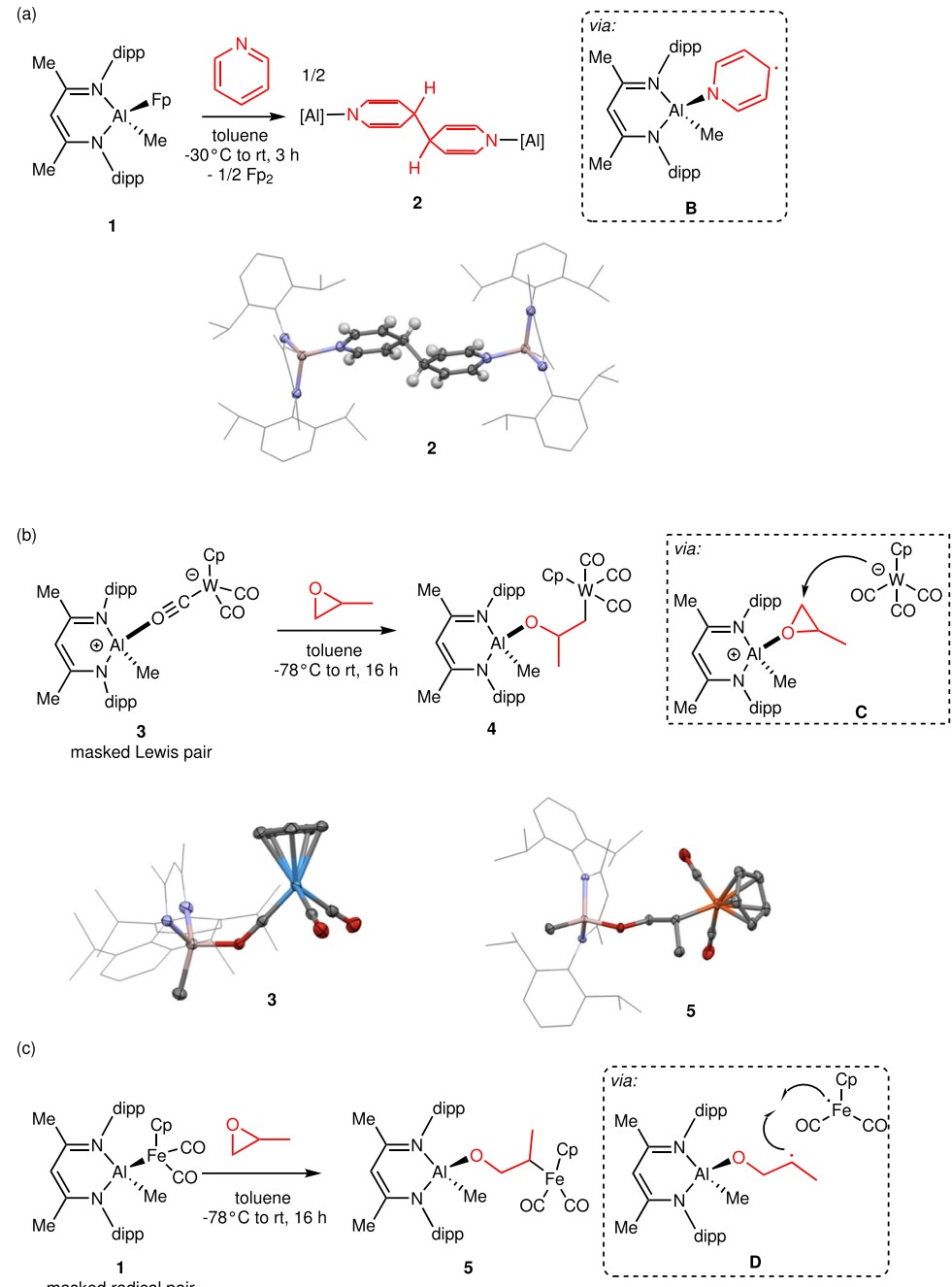

**Fig. 2 | Experimental evidence for Al/Fe frustrated radical pair (FRP) behavior.**
**a** Pyridine diradical coupling induced by coordination to [LAlMe]•, (**b**, **c**) divergent regioselectivity during ring opening of (±)-propylene oxide consistent with **3** acting as a masked frustrated Lewis pair (FLP) and **1** acting as a masked frustrated radical pair (FRP). Crystal structures are shown as thermal ellipsoids (50% probability) for key atoms and wireframes for C atoms in L. Pyridine H atoms are shown in calculated positions, and all other H atoms are omitted for clarify.

NMR spectroscopy showed the formation of LAl(Me)X products **6** and stoichiometric FpH. Formation of FpH was evident in these experiments from the appearance of resonances at 4.1 ppm (Cp) and −11.7 ppm (Fe-H) in the [1]H NMR spectra (see Supplementary Fig. 22)[36].

To gain further insight into the X-H activation mechanism, we measured pseudo-first order rate constants (excess X-H, 283 K) as a function of X. Effectively no variation in reaction rate was observed across the series, nor was a kinetic isotope effect (KIE) evident when comparing MeOH and MeOD (Fig. 3b). The fact that X-H activation rate is independent of its p$K_a$ (e.g. [i]PrOH vs. [i]BuNH$_2$) is inconsistent with the X-H cleavage involving simple H[+] transfer and, instead, implies that e[-] transfer is involved in limiting the rate. The fact that X-H cleavage occurs independently of steric hindrance (e.g. MeOH vs. [t]BuOH) implies that the rate-limiting step does not involve substrate coordination. The absence of a measurable KIE indicates that X-H activation occurs after the rate-limiting step. Collecting these observations, we propose that the rate-limiting step in each of the reactions is Al-Fe cleavage from **1** to reveal the **A**•/Fp• FRP. Both CIBW of the X-H substrate and subsequent X-H cleavage by PCET would, then, occur after rate-determining formation of intermediate **A**•. Consistent with the fact that masked FLP **3** proceeds by a different mechanism than masked FRP **1**, we found that X-H cleavage by **3** does have a p$K_a$ dependence. For example, complex **3** was found to activate more acidic [i]PrOH but not less acidic [i]PrNH$_2$.

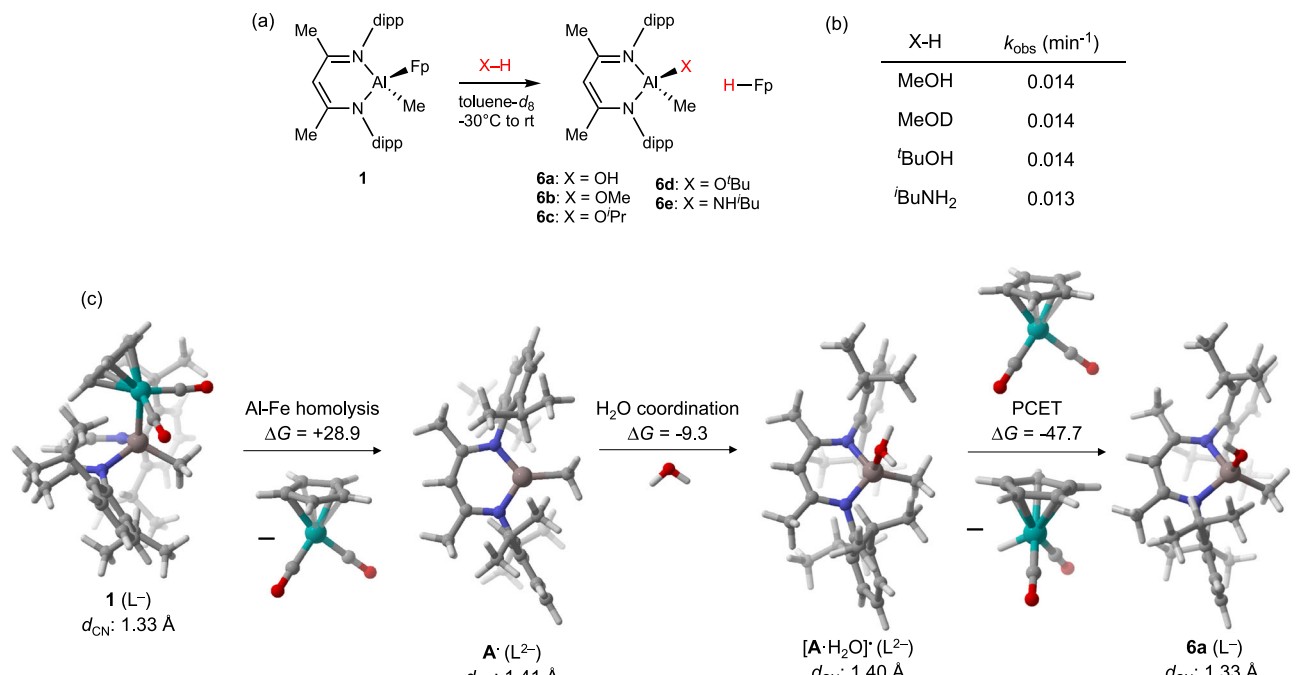

**Fig. 3 | Examination of X-H activation pathways. a** Experimental X-H bond activation reactions, (**b**) pseudo-first order rate constants (excess X-H, 283 K) determined by ¹H NMR spectroscopy, (**c**) reaction thermodynamics computed by DFT. Gibbs free energy values are given in units of kcal/mol, $d_{CN}$ values are average C-N distances for each compound, and error bars (95% confidence intervals) on $k_{obs}$ are roughly ± 0.002 min⁻¹.

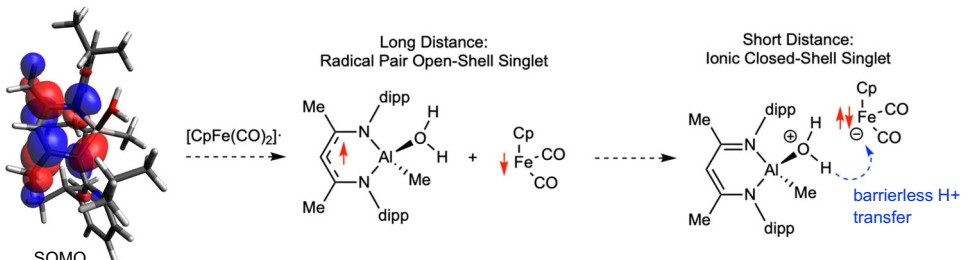

**Fig. 4 | Asynchronous proton-coupled electron transfer (PCET).** Singly-occupied molecular orbital for [**A**·H₂O]˙ (0.03 isosurface) calculated by DFT (left), and a summary of lowest-energy electronic configurations as a function of [Al-OH₂]···Fe distance (right). The compiled data indicate asynchronous PCET in which ET leads PT.

To examine the viability of our mechanistic proposal, we used unrestricted DFT calculations (M06/def2-TZVPD//PBE1PBE/def2-SVP in toluene CPCM solvent, see Supplementary Information for details) to model key reaction steps and intermediates for H₂O activation by **1** (Fig. 3c). As we calculated previously, homolytic Al-Fe cleavage of **1** to produce the **A**˙/Fp˙ FRP is endergonic (ΔG = +28.9 kcal/mol)[22]. Coordination of H₂O to **A**˙ to produce [**A**·H₂O]˙ was calculated to be slightly exergonic (ΔG = −9.3 kcal/mol). Finally, net H-atom transfer from [**A**·H₂O]˙ to Fp˙ to produce **6a** and FpH was calculated to be highly exergonic (ΔG = −47.7 kcal/mol). Thus, the overall H₂O activation process is thermodynamically favorable by approximately ΔG = −28.1 kcal/mol. The calculated endergonic Al-Fe cleavage step and subsequent exergonic steps for water coordination and H-atom transfer is consistent with experimental observations indicating that both H₂O coordination and O-H bond cleavage occurring after the rate-determining step. It is noteworthy that the energy profile of H₂O activation is distinct from that previously calculated for CO₂ activation in that CO₂ coordination to **A**˙ was calculated to be slightly endergonic[22]. Thus, pseudo-first order rate constants previously measured[22] for CO₂ activation were likely composites of the two elementary rate constants for Al-Fe cleavage and CO₂ coordination. On the other hand, the pseudo-first order rate constants for X-H activation in this study are more likely pure measurements of Al-Fe homolysis to unmask the key FRP intermediate.

Towards understanding the origin of CIBW in this system, next we examined the electronic configurations of key intermediates. In starting complex **1**, the β-diketiminate ligand is in its closed-shell [L]⁻ form and has correspondingly short C-N distances indicative of significant C=N double-bond character (Fig. 3c). Al-Fe homolysis produces intermediate **A**˙ in which the β-diketiminate ligand is reduced to its [L˙]²⁻ form, with elongated C-N distances due to population of the ligand π* manifold with additional electron density. This reduced ligand character is maintained upon H₂O coordination to Al^III to form [**A**·H₂O]˙ according to the calculated C-N distances. Furthermore, in [**A**·H₂O]˙, the unpaired electron is localized on the β-diketiminate ligand's π* system rather than at Al according to examination of its SOMO (Fig. 4) and spin density distribution. The electronic configuration of the [**A**·H₂O]˙/Fp˙ radical pair was found to depend on interaction distance. At long distances (>4 Å), the lowest-energy configuration of the [**A**·H₂O]˙/Fp˙ pair is a weakly coupled, open-shell singlet diradical ($S^2 > 0$) where one electron remains in the β-diketiminate π* system and the other electron is localized on the Fe center. The triplet state is

several kcal/mol higher in energy. At shorter interaction distances (<4 Å), it is thermodynamically favorable for e⁻ transfer to occur such that the β-diketiminate π* electron is transferred to the Fe center to generate Fp⁻ and closed-shell [LAl(Me)(OH₂)]⁺. Populating this polar electronic configuration induces barrierless proton transfer (Fig. 4). Upon net hydrogen atom transfer to Fp•, the contracted C-N distances calculated for **6a** imply re-oxidation to the [L]⁻ state. Thus, perhaps the best description of this PCET step is that it asynchronously couples H⁺ transfer from acidified H₂O to e⁻ transfer from the β-diketiminate electron reservoir. However, though these results imply that e⁻ transfer precedes H⁺ transfer, the available data cannot elucidate the degree of coupled timing for e⁻ transfer and H⁺ transfer along the asynchronous PCET pathway. It is worth contrasting this behavior with a recently reported computational model[26] for X-H cleavage by a heterobinuclear Al-M complexes[24] with diradical character[25]. While facile X-H cleavage processes were observed in that system, they are proposed to involve a concerted, 2e⁻ pathways and thus do not strictly qualify as PCET reactions enabled by CIBW.

Finally, we calculated the degree of CIBW by evaluating the BDFE$_{O-H}$ for [**A**·H₂O]•. DFT calculations with the M06 functional gave a value of only 1.0 kcal/mol, and other DFT functionals (e.g. PBE1PBE) gave similarly small BDFE value. CCSD(T)/def2-TZVP energies on a model complex confirmed this very low bond energy for the O-H bond in [**A**·H₂O]•. These results imply that the BDFE$_{O-H}$ for [**A**·H₂O]• is significantly lower than that of [Sm(OH₂)ₙ]²⁺[17,18], meaning that the degree of CIBW induced by **A**• in the current system is without precedent. Moreover, the near-zero BDFE$_{O-H}$ for [**A**·H₂O]• raises another mechanistic possibility: upon H₂O coordination to **A**•, [**A**·H₂O]• can spontaneously liberate H• to form **6a** directly without any interaction with Fp•. This hypothetical step can also be considered PCET, as it would involve loss of H⁺ from coordinated H₂O coupled to loss of e⁻ from the reduced β-diketiminate ligand. This alternative pathway, which is somewhat analogous to C-O cleavage induced by epoxide coordination to **A**• (Fig. 2c), cannot be ruled out. In fact, if Al-Fe homolysis from **1** produced a solvent-caged **A**•/Fp• FRP, it is possible that both pathways (asynchronous PCET from [**A**·H₂O]• to Fp• and direct H• liberation from [**A**·H₂O]• followed by H•/Fp• recombination) are operative. Whereas DFT calculations indicate that the coordinated water molecule in [**A**·H₂O]• is not acidified significantly compared to free H₂O, the degree of CIBW in the current system is driven mainly by the strong reducing potential of the 1e⁻-reduced β-diketiminate ligand in [**A**·H₂O]• (see SI for detailed calculations).

## Discussion

Homolysis at ambient conditions of Al-Fe complex **1** produces small equilibrium concentrations of a frustrated radical pair (FRP) consisting of Fp• and the Alᴵᴵᴵ-containing radical species **A**•. Previously established coordination-induced bond weakening (CIBW) of C=O π-bonds by **A**• is now shown definitively to extend to C-O σ-bonds, representing a metal/ligand cooperative analogue of established Tiᴵᴵᴵ chemistry[37]. This FRP behavior has been applied to CIBW of O-H and N-H σ-bonds. Experimental and computational analysis of these X-H activation reactions indicate that **A**• induces a significant degree of CIBW. Not only is this a rare example of CIBW by aluminum, the most abundant metal on earth, but it establishes a metal/ligand cooperative paradigm for multisite PCET wherein the metal center acidifies a coordinated X-H bond while the redox non-innocent ligand conducts e⁻ transfer in a concerted but asynchronous manner. The use of earth-abundant metals in PCET reactions will be critical to various renewable energy schemes. The results reported here will inform the molecular design features of such systems, opening CIBW to abundant but redox inert metals through metal/ligand cooperativity.

## Methods

Experimental details (synthesis & characterization, reactivity studies, kinetics measurements) are provided as Supplementary Information.

Computations were performed as follows. The PBE1PBE[38] functional (ultrafine integration grid), def2-SVP basis set[39], and conductor-like polarizable continuum model (CPCM)[40] for toluene were used for geometry optimizations and vibrational frequency characterization in Gaussian 16[41]. Single-point energies were calculated with M06[42]/def2-TZVPD using ORCA[43]. CCSD(T) calculations were executed in ORCA. 3D structures were created using CYLview[44]. Further computational considerations are provided as Supplementary Information.

## Data availability

All data generated or analyzed during this study are included in this published article and its supplementary information files or by download from the Cambridge Crystallography Data Center (CCDC deposition numbers 2297603-2297605). Coordinates of computationally optimized structures are provided as source data. All other data are available from the corresponding author upon request. Source data are provided with this paper.

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

## Acknowledgements

This material is based upon work supported by the U.S. Department of Energy (DOE), Office of Science, Office of Basic Energy Sciences under Award Number DE-SC0021055 to N.P.M and by the US National Science Foundation with award CHE-2153215 to D.H.E. X-ray crystallography data for complex **2** were obtained at NSF's ChemMatCARS Sector 15, which is supported by the Divisions of Chemistry (CHE) and Materials Research (DMR), National Science Foundation, under grant number NSF/CHE-1834750. Use of the Advanced Photon Source, an Office of Science User Facility operated for the DOE Office of Science by Argonne National Laboratory, was supported by the DOE under Contract No. DE-AC02-06CH11357. Computational resources were provided by the Advanced Cyberinfrastructure for Education and Research (ACER) group at UIC and the Office of Research Computing at BYU. Dr. Daniel McElheny (UIC) assisted with NMR spectroscopy. This manuscript is dedicated to the memory of Prof. Jeffrey A. Byers, an inspiring scientist and friend who provided valuable suggestions related to this study.

## Author contributions

S.S. and R.P.S. performed experimental studies. M.R.R. and J.K. performed computational studies. N.P.M. and D.H.E. supervised the work.

## Competing interests

The authors declare no competing interests.

## Additional information

**Peer review information** : *Nature Communications* thanks the anonymous reviewers for their contribution to the peer review of this work. A peer review file is available.

