## [Peer Review File · Nature Communications]

Coordination-induced O-H/N-H bond weakening by a redox non-innocent, aluminum-containing radicalReviewers' Comments:

Reviewer #1:

Remarks to the Author:

Mankad et al. report an experimental investigation on the coordination-induced O-H and N-H bond weakening using a previously studied heterobinuclear [LAl(Me)Fp] complex 1, L = [HC(CMeNdipp)₂]-, dipp = 2,6-di-iso-propylphenyl, Fp = [FeCp(CO)₂]-. The work follows on other studies of the same authors on the same system, where they provide experimental and computational evidence that complex 1 dissociates reversibly at ambient conditions by Al-Fe homolysis, producing small equilibrium concentrations of the [LAlMe]. and Fp. frustrated radical pair (FRP) that can cooperatively activate oxygenated substrates such as CO₂. Here, the authors show stoichiometric activation of O-H and N-H containing substrates including H₂O, MeOH, iPrOH, tBuOH as well as iBuNH₂. Based on NMR and kinetic measures, they propose that the rate-determining step (RDS) of the reaction mechanism is Al-Fe cleavage from 1 to produce [LAlMe]. / Fp. FRP followed by substrate (H₂O) coordination to [LAlMe]. to give [LAlMeH₂O]., from which H-atom transfer to Fp. occurs via H⁺/e⁻ transfer from the redox non innocent ligand (PCET).

The present work provides a clear experimental evidence of coordination-induced O-H and N-H bond weakening, which is an outstanding result. However, the nature and reactivity of this class of heterobinuclear complexes is not well-contextualized in the Introduction, since references to previously studied metal-alumanyl systems which have experimentally shown reactivity with small molecules are lacking (<https://doi.org/10.1038/s41557-018-0198-1>). Metal-alumanyl systems have been also thoroughly characterized computationally (<https://doi.org/10.1021/jacs.1c06728>) to the extent that, very recently, computational studies have demonstrated that O-H and N-H bond activation could be both kinetically and thermodynamically feasible, through a similar diradical-like mechanism (<https://doi.org/10.1002/chem.202203584>). Within this broader context, the conclusion of this work:..."this is the first example of coordination-induced bond weakening by aluminum"... is questionable.

The computational part needs to be improved considerably.

1) DFT calculations have been performed only for modeling the reaction intermediates in the proposed mechanism for H₂O activation by 1, thus reporting only thermodynamic free energies and NOT kinetic free energy barriers, which are fundamental for identifying the rate-determining step. Indeed, the assumed RDS (homolytic Al-Fe cleavage of 1) is rather highly endothermic ($\Delta G = +24.5$ kcal/mol), which means that the activation barrier should be expected to be even much higher (that could be not completely consistent with a reaction occurring at a temperature from -30°C to room temperature). A transition state for the PCET (3TS) has been located at a lower level of theory (which is then not consistently comparable to intermediates) on the triplet energy surface. I would expect here a spin-forbidden reaction, where a spin-crossing from a singlet to a triplet PES should occur. Therefore, calculations on both the PESs would be needed to properly characterize the mechanism (at least to verify that the TS energy is lower on the triplet than on the singlet PES) and confirm the asynchronicity of the PCET process.

2) In the BDFE calculation for eq 2, an error of about 14 kcal/mol has been assessed with respect to the experimental value (63 kcal/mol), which suggests that the used computational set up is not sufficiently reliable (for instance, relativistic effects were not included, the basis set quality and functionals other than PBE0 were not tested).

3) It would be helpful to present the electronic structures or, at least, the calculated spin density of the radical species to show where the unpaired electron is (de)localized in the intermediates. The authors claim that the β -diketiminato ligand is redox non-innocent and that upon reduction the extra electron populate the ligand n^* manifold. Did the authors calculate the electronic structure? Changes in intraligand bond distances cannot be used as indicative of the ligand oxidation state. At DFT level,

oxidation state is a very critical issue.

4) It may be useful a comparative study with recent computational results showing that the M-Al bond is able to activate, via a concerted, diradical-like mechanism, the O-H and N-H bonds. Moreover, analogy of complex 1 with CaMn₄ oxygen-evolving complex in photosystem-II should be more deeply justified.

Although the reported experimental evidence is very important in the field of small molecule activation processes, based on the above critical issues, I am afraid that I cannot recommend the acceptance of this manuscript in Nature Communications as it stands.

Reviewer #2:

Remarks to the Author:

See attached review

Mankad and coworkers present an interesting example of multisite proton coupled electron transfer that occurs through the homolysis of an Fe-Al bond that results in an Al(III) complexed to a radical anion ligand that induces significant bond-weakening in small molecules coordinated to the Al. Overall, the work is quite interesting and the mechanistic work and analysis supports the mechanistic hypotheses in the manuscript. I am strongly in favor of acceptance, but have a number of comments for the authors to consider in advance of publication.

Some of the description and citations in the introduction neglect seminal work. I recommend that either seminal work be cited in place of the work cited in the manuscript, or that it be added.

In the section describing the frontier work in PCET in synthesis, the citation of the recent *Chem. Rev.* by Knowles and colleagues is highly relevant, but I do not believe the work of Studer is appropriate. While certainly photocatalytic, the reaction requires the synthesis of a high MW sacrificial phosphine that limits the utility of the system. Earlier work by Knowles (*J. Am. Chem. Soc.*, 2015, **137**, 6440–6443) is really more appropriate to cite.

I am not sure that I agree with the description of the basis for coordination induced bond weakening, especially the use of invariably. While in most instances coordination of a small molecule to a low-valent metal metal reduces the pKa of the bound of the bound ligand, in some cases it is quite modest. In other cases, significant X-H bond-weakening occurs at less proximate sites although the impact of increasing acidity is almost non-existent (ie coordination of amides to Ti(III), Sm(II), etc.)

Sm-water bond-weakening was established by Flowers several years earlier (2015) than the cited Mayer report (*J. Am. Chem. Soc.* **2015**, *137*, 11526-11531). The estimate of bond-weakening by Flowers described in a follow-up small review (*Dalton Trans.* **2019**, *48*, 16142-16147) is smaller but consistent with the value determined by Mayer and Kolmar employing thermochemical cycles but is more relevant to synthetic systems in organic media. The value by Mayer is an overestimate since aqueous potentials and pKa's were used not relevant to the solvent employed in the reduction of an enamine in THF. I believe this was also recently pointed out by Peters and coworkers in a recent *JACS* publication as well. This should be corrected in the introduction and in the results and discussion section below Figure 3. It isn't necessary to get into this level of detail, but certainly the range of bond weakening based on *experimental* evidence can be described briefly.

In Figure 1 a, the incorrect structure is shown. Upon addition of water to low-valent titanocene, chloride ions are solvated by water and displaced to the outer sphere. A better representation

n = 1,2

is shown above and supported by EPR, voltammetry, and computational studies carried out by Gansauer and coworkers (*Angew. Chem. Int. Ed.*, 2012, **51**, 3266–3270). I recommend this structure be shown.

The discussion around Figure 1b should be referred to as multisite PCET. This terminology is well-established for the description of biological systems such as the OEC of photosystem II and synthetic systems. 1b is also a classic example of acidification of a bound small molecule coupled to ET from a redox center in the language used to describe coordination induced bond weakening.

In the second paragraph of the results and discussion, it may be useful to cite recent work of Knowles on the coordination induced bond weakening of cyclopropanes (*J. Am. Chem. Soc.* 2022, 144, 34, 15488–15496) since in some ways it is analogy to the current system.

In the weakening of N-H bonds, recent work has demonstrated significant weakening in Sm(II)-NH₃ N-H bonds (*J. Org. Chem.* **2022**, 87, 1689-1697).

In the conclusion, the authors discuss the extension to the weakening of C-O bonds. I think this should be reworded. In the reduction of epoxides by low-valent titanocene, coordination induced bond weakening is an important feature of the first step in reduction. Although it doesn't involve X-H bond weakening, coordination of an epoxide to Ti(III) clearly weakens the C-O bond and is a classic example of coordination induced bond weakening. There needs to be more context in the description of the present Al-Fe system to differentiate from more classic systems.

Reviewer #3:

Remarks to the Author:

Coordination-induced O–H/N–H bond weakening by a redox non-innocent, aluminium-containing radical.

Mankad and coworkers report a remarkable discovery based on coordination-induced bond weakening of protic substrates on coordination to a aluminium radical. These radical is derived from the homolysis of a Fe–Al heterometallic. The authors have provided convincing evidence not only for radical generation (which expands and builds upon prior work from the group, e.g J. Am. Chem. Soc. 2022, 3210) but also that a PCET event is at play, rather than simply an acidification of O–H or N–H bond on coordination.

The results are important and will find broad interest as they suggest a general strategy for use of H₂O, NH₃, simple alcohols and amines through activation at main-group radical intermediates. Ultimately the information could inform design of new catalysts. The authors are also correct in stressing then key point of novelty that this is the first time such reactivity has been observed at aluminium, the most abundant metal in the earth's crust.

I am very supportive of publication and have the following suggestions.

(1) The narrative of the current draft is a little awkward. The work is framed in terms of PCET but the bulk of the initial results and discussion, captured in Figure 2, focuses on reactivity of heterometallics. At times this feels likely two stories. Can the authors edit this section to reduce the amount of content, or better draw the connection to the PCET results – which are the key novelty.

(2) Additional analysis of the electronic structure of A by DFT, including a spin-density plot would help argue the ligand-based character of the radical.

(3) The estimation of the BDE of [A–H₂O] radical is a little unsatisfying. I think it is appropriate to put an upper-bound on this, but the lack of correlation between the computational values and experimental data in benchmarking of model is a bit worrisome. The use of diffuse functions in the basis-set is known to be important to properly model metal hydride complexes, I would strongly suggest that the authors investigate Ahlrich's basis set that includes both polarisation and diffuse e.g. def2-QZVPPD or def2-TZVPD and compare data to those in Table S1. The functional group dependence should also be explored including both hybrid and mGGA functionals.

(4) It is not uncommon to not be able to observe ¹³C resonance of sites directly bonded to Al by 1D NMR, but usually these are resolved by 2D methods. Can the authors find these for 6a-e by 1H-¹³C HSQC experiments.

(5) Can 183W satellite peaks not be observed in the NMR data of 4? If not a note should be added to the SI to clarify.

(6) There is a serious issue with the characterisation data reported for 6d. This compound does not contain W, something is wrong here. Please check the data and assignments.

Reviewer #4:

Remarks to the Author:

The structural data presented by the authors does support their conclusion, but RS15 and SS1 need refining with sensible cutoff on the data resolution, see comments below. For compound 2 the description in the manuscript of the numbering and bond distances are not consistent with the CIF, I think the atom numbers were changed at somepoint, see below. Other standard information is missing from the CIFs that the authors need to fill in.

RS15

While the data was collected to 0.55 angstroms, data is considered observed is $I/\sigma(I) > 2$, see INTENSITY STATISTICS FOR DATASET below. Therefore from ~0.72 to 0.55 angstroms the data is on

average unobserved. This mean a massive amount of noise was added to this data set. If the data are cut at 0.72 Angstroms then the high electron density peaks around the W drop from ~4 electrons to ~1.5 electrons. In cutting the data all the AlertA and B's are sorted out and do not occur and leading to a better refined structure.

Why are there no hydrogen atoms on the Cyclopentadienyl ligands? Why are its hydrogen not included in the chemical formula?

Please fill out "_exptl_crystal_colour?" in the CIF even if the crystal is colourless.

INTENSITY STATISTICS FOR DATASET # 1 rs15.hkl

Resolution #Data #Theory %Complete Redundancy Mean I Mean I/s Rmerge Rsigma

Inf	-	2.29	364	374	97.3	6.34	1156.36	50.20	0.0245	0.0124
2.29	-	1.52	850	850	100.0	7.03	752.48	29.89	0.0377	0.0205
1.52	-	1.20	1221	1221	100.0	7.09	480.20	19.41	0.0592	0.0337
1.20	-	1.04	1261	1261	100.0	7.05	347.86	13.89	0.0825	0.0492
1.04	-	0.95	1130	1130	100.0	6.96	260.09	10.11	0.1162	0.0692
0.95	-	0.88	1208	1208	100.0	6.82	216.10	7.99	0.1480	0.0902
0.88	-	0.82	1389	1389	100.0	6.52	167.55	5.78	0.2108	0.1253
0.82	-	0.78	1188	1188	100.0	6.30	138.36	4.51	0.2621	0.1594
0.78	-	0.75	1052	1052	100.0	6.09	110.42	3.64	0.3239	0.2052
0.75	-	0.72	1235	1235	100.0	5.92	92.28	3.03	0.3796	0.2519
0.72	-	0.69	1451	1452	99.9	5.67	68.85	2.23	0.4645	0.3454
0.69	-	0.67	1126	1127	99.9	5.49	53.15	1.71	0.5702	0.4593
0.67	-	0.65	1245	1245	100.0	5.25	46.08	1.46	0.6726	0.5487
0.65	-	0.63	1440	1440	100.0	5.04	41.91	1.23	0.7734	0.6375
0.63	-	0.62	772	773	99.9	4.85	35.11	1.03	0.8179	0.7731
0.62	-	0.60	1728	1729	99.9	4.79	20.72	0.57	0.9493	1.3281
0.60	-	0.59	959	959	100.0	4.63	13.79	0.34	1.0196	2.0491
0.59	-	0.58	1008	1008	100.0	4.49	3.30	0.02	1.2293	8.8490
0.58	-	0.56	2244	2244	100.0	4.30	2.38	-0.03	1.2744	13.0452
0.56	-	0.55	1262	1264	99.8	4.09	0.81	-0.09	1.3371	40.6408

RS06_m

b cell parameter is missing a decimal place, give as 18.481 but error is given as 0.0003 why?

Please add the colour and size of the crystal into the CIF.

Please label the atoms consistently with the other structures and not just use the number straight out of SHELXT.

SS1

While the data was collected to 0.50 angstroms, data is considered observed is $I/\sigma(I) > 2$, see INTENSITY STATISTICS FOR DATASET below. Therefore from ~0.71 to 0.50 angstroms the data is on average unobserved. This mean a massive amount of noise was added to this data set. Also R(merge) even at 0.71 angstroms is 63.96% which again is adding more noise to the data. From my experience R(merge)s of greater than 30% stop adding more noise than useful data. The authors may want to go back at look at the processing of the original data and determine if the crystal was dying in the beam and if possible cut the data at that point. However, if the data were cut at 0.82 Angstrom all the AlertA and B's are no longer triggered.

Was the temperature of the data collection really 273K?

Please add crystal dimension into the CIF.

Please add the following into the CIF

_diffrn_source 'Advanced Photon Source, NSFs ChemMatCARS'

Please number this atom consistently with those of the other structures C00Q

Please check:

"The X-ray crystal structure of 2 indicates localized π -bonding consistent with disruption of pyridine aromaticity: the C2-C3 and C5-C6 distances show double-bond character [1.341(2)-1.343(2) Å] while the C3-C4 and C4-C5 distances show single-bond character [1.510(2)-1.516(3) Å], and the N-C2 and N-C6 distances are elongated [1.396(2)-1.400(2) Å] compared to pyridine (1.340 Å)²⁶. The C4-C4' distance is also indicative of single bonding [1.569(3) Å], consistent with the pyramidalized, C(sp³)-like geometries at these centers."

The bonds listed above are not consistent with the number in the CIF provided, as I believe these should be in the thirties, C2-C3 are in the backbone of the main ligand.

INTENSITY STATISTICS FOR DATASET # 1 SS1.hkl

Resolution #Data #Theory %Complete Redundancy Mean I Mean I/s Rmerge Rsigma

Inf - 2.07	406	408	99.5	11.39	598.68	12.99	0.0613	0.0657
2.07 - 1.37	969	969	100.0	12.86	184.89	11.78	0.1148	0.0653
1.37 - 1.08	1386	1386	100.0	12.83	105.41	9.85	0.1623	0.0730
1.08 - 0.94	1397	1397	100.0	13.03	46.74	6.98	0.2601	0.1020
0.94 - 0.86	1266	1266	100.0	12.37	27.57	4.93	0.3674	0.1542
0.86 - 0.79	1523	1523	100.0	13.12	19.94	4.00	0.4659	0.2038
0.79 - 0.75	1172	1172	100.0	13.31	13.58	2.91	0.5893	0.2981
0.75 - 0.71	1448	1448	100.0	12.20	12.03	2.34	0.6396	0.3757
0.71 - 0.68	1297	1301	99.7	11.94	9.42	1.80	0.7118	0.5134
0.68 - 0.65	1529	1544	99.0	11.54	7.70	1.42	0.8000	0.6748
0.65 - 0.63	1209	1214	99.6	9.80	5.66	0.94	0.8909	1.0273
0.63 - 0.61	1377	1384	99.5	8.75	4.71	0.73	0.9446	1.3584
0.61 - 0.59	1545	1559	99.1	8.00	4.38	0.64	0.9467	1.5889
0.59 - 0.57	1783	1796	99.3	7.28	3.19	0.43	1.0328	2.3717
0.57 - 0.56	964	976	98.8	6.13	2.57	0.30	1.0808	3.3025
0.56 - 0.55	1024	1037	98.7	5.49	1.32	0.14	1.1668	6.8540
0.55 - 0.53	2373	2401	98.8	5.86	1.02	0.10	1.2159	8.9865
0.53 - 0.52	1306	1329	98.3	5.72	1.22	0.12	1.1832	8.0135
0.52 - 0.51	1403	1421	98.7	5.72	0.72	0.07	1.2187	14.0790
0.51 - 0.50	1548	1583	97.8	5.63	0.46	0.02	1.2779	23.2705

In addition to changes indicated below in response to reviewer comments, the length of the abstract was cut down to fall within Nat. Commun. regulations, and several typos were fixed. All new text in the manuscript is highlighted in yellow.

Reviewer #1 (Remarks to the Author):

Mankad et al. report an experimental investigation on the coordination-induced O-H and N-H bond weakening using a previously studied heterobinuclear [LAl(Me)Fp] complex 1, L = [HC(CMeNdipp)₂], dipp = 2,6-di-iso-propylphenyl, Fp = [FeCp(CO)₂]. The work follows on other studies of the same authors on the same system, where they provide experimental and computational evidence that complex 1 dissociates reversibly at ambient conditions by Al-Fe homolysis, producing small equilibrium concentrations of the [LAlMe]. and Fp. frustrated radical pair (FRP) that can cooperatively activate oxygenated substrates such as CO₂. Here, the authors show stoichiometric activation of O-H and N-H containing substrates including H₂O, MeOH, iPrOH, tBuOH as well as iBuNH₂. Based on NMR and kinetic measures, they propose that the rate-determining step (RDS) of the reaction mechanism is Al-Fe cleavage from 1 to produce [LAlMe]. / Fp. FRP followed by substrate (H₂O) coordination to [LAlMe]. to give [LAlMeH₂O]., from which H-atom transfer to Fp. occurs via H⁺/e⁻ transfer from the redox non innocent ligand (PCET).

The present work provides a clear experimental evidence of coordination-induced O-H and N-H bond weakening, which is an outstanding result. However, the nature and reactivity of this class of heterobinuclear complexes is not well-contextualized in the Introduction, since references to previously studied metal-alumanyl systems which have experimentally shown reactivity with small molecules are lacking (<https://doi.org/10.1038/s41557-018-0198-1>). Metal-alumanyl systems have been also thoroughly characterized computationally (<https://doi.org/10.1021/jacs.1c06728>) to the extent that, very recently, computational studies have demonstrated that O-H and N-H bond activation could be both kinetically and thermodynamically feasible, through a similar diradical-like mechanism (<https://doi.org/10.1002/chem.202203584>). Within this broader context, the conclusion of this work:..."this is the first example of coordination-induced bond weakening by aluminum"..., is questionable.

Response: Thank you to Reviewer 1 for pointing out these omissions. We have made edits to the wording in the Introduction paragraph about CIBW by aluminum and added these three references.

The computational part needs to be improved considerably.

1) DFT calculations have been performed only for modeling the reaction intermediates in the proposed mechanism for H₂O activation by 1, thus reporting only thermodynamic free energies and NOT kinetic free energy barriers, which are fundamental for identifying the rate-determining step. Indeed, the assumed RDS (homolytic Al-Fe cleavage of 1) is rather highly endothermic ($\Delta G = +24.5$ kcal/mol), which means that the activation barrier should be expected to be even much higher (that could be not completely consistent with a reaction occurring at a temperature from -30°C to room temperature). A transition state for the PCET (3TS) has been located at a lower level of theory (which is then not consistently comparable to intermediates) on the triplet energy surface. I would expect here a spin-forbidden reaction, where a spin-crossing from a singlet to a triplet PES should occur. Therefore, calculations on both the PESs would be needed to properly characterize the mechanism (at least to verify that the TS energy is lower on the triplet than on the singlet PES) and confirm the asynchronicity of the PCET process.

Response: In collaboration with the Ess group (a computational chemistry group), we have now thoroughly examined the energy surface for reaction steps that can occur after Al-Fe bond homolysis.

We confirmed that charge separated species (bond heterolysis) are ~60 kcal/mol higher in energy (consistent with nonpolar, non-hydrogen bonding solvent) than Al and Fe radicals (bond homolysis). Also, as expected, there is no potential energy barrier for bond homolysis. There is the normal, smooth dissociation. While there could be a variational transition state in principle, finding this would be an enormous effort and would likely show that the barrier for bond cleavage is within the error of DFT compared to the calculated bond energy. Therefore, the bond energy is likely a reasonable, but not perfect, estimate of the kinetics for this bond cleavage. Unfortunately, DFT likely overestimates this bond energy. While it would be nice to calculate it with CCSD(T), the full system is needed since the bulky aryl groups greatly modulate the bond energy, making this system is too large even for modern compute systems (we tried; even with ORCA's fast solver it was too large).

As suspected by the reviewer, long-range interaction of the Al and Fe radicals has a lower energy open-shell singlet configuration, not a triplet configuration. The triplet state is not involved in the reaction. Orbital swapping was required to obtain the correct open-shell singlet configuration, and we confirmed the configuration with a CASSCF calculation. Importantly, when water is coordinated to the Al metal center to generate Al(H₂O) and the Fe center can approach the proton of water, at a distance of ~4 Angstroms electron transfer becomes thermodynamically favorable. The electron transfer happens from the Al ligand to the Fe metal center, and this results in a switch from an open-shell singlet configuration to a closed-shell singlet configuration; this induces barrierless proton transfer. A detailed discussion of these new calculations is now contained in the manuscript.

2) In the BDFE calculation for eq 2, an error of about 14 kcal/mol has been assessed with respect to the experimental value (63 kcal/mol), which suggests that the used computational set up is not sufficiently reliable (for instance, relativistic effects were not included, the basis set quality and functionals other than PBE0 were not tested).

Response: With the new understanding about the reaction thermodynamics and electronic states it was no longer necessary to invoke eq 1-3. We have now removed the section discussing these equations and instead added an entirely new paragraph describing the direct calculation of the Al-water O-H BDFE. We used both DFT (M06, PBE1PBE) and CCSD(T) calculations to verify this very small bond energy.

3) It would be helpful to present the electronic structures or, at least, the calculated spin density of the radical species to show where the unpaired electron is (de)localized in the intermediates. The authors claim that the β -diketiminato ligand is redox non-innocent and that upon reduction the extra electron populate the ligand π^* manifold. Did the authors calculate the electronic structure? Changes in intraligand bond distances cannot be used as indicative of the ligand oxidation state. At DFT level, oxidation state is a very critical issue.

Response: Thank you, this is now shown in revised Figure 4. The unpaired electron is, indeed, completely localized on the ligand backbone.

4) It may be useful a comparative study with recent computational results showing that the M-Al bond is able to activate, via a concerted, diradical-like mechanism, the O-H and N-H bonds. Moreover, analogy of complex 1 with CaMn₄ oxygen-evolving complex in photosystem-II should be more deeply justified.

Response: We added the following text to contrast our report with the previous work on the Al-Au complexes: "It is worth contrasting this behavior with a recently reported computational model for X-H cleavage by a heterobinuclear Al-M complexes with diradical character. While facile X-H cleavage processes were observed in that system, they are proposed to involve a concerted, 2e- pathways and thus do not strictly qualify as PCET reactions enabled by CIBW." As suggested by Reviewer 2, we have included the established terminology of "multisite PCET" to describe both the CaMn₄ OEC and our

reported system. We hope this further justifies the analogy. We have also removed mention of the OEC in the Results & Discussion section to avoid overemphasizing this analogy.

Although the reported experimental evidence is very important in the field of small molecule activation processes, based on the above critical issues, I am afraid that I cannot recommend the acceptance of this manuscript in Nature Communications as it stands.

Response: We thank Reviewer 1 for the thorough and rigorous review. After incorporation of revisions in response to these comments and requests from other reviewers, we are confident the manuscript will be acceptable for Nat. Commun.

Reviewer #2 (Remarks to the Author):

Mankad and coworkers present an interesting example of multisite proton coupled electron transfer that occurs through the homolysis of an Fe-Al bond that results in an Al(III) complexed to a radical anion ligand that induces significant bond-weakening in small molecules coordinated to the Al. Overall, the work is quite interesting and the mechanistic work and analysis supports the mechanistic hypotheses in the manuscript. I am strongly in favor of acceptance, but have a number of comments for the authors to consider in advance of publication.

Response: Thank you to Reviewer 2 for the supportive evaluation.

Some of the description and citations in the introduction neglect seminal work. I recommend that either seminal work be cited in place of the work cited in the manuscript, or that it be added. In the section describing the frontier work in PCET in synthesis, the citation of the recent *Chem. Rev.* by Knowles and colleagues is highly relevant, but I do not believe the work of Studer is appropriate. While certainly photocatalytic, the reaction requires the synthesis of a high MW sacrificial phosphine that limits the utility of the system. Earlier work by Knowles (*J. Am. Chem. Soc.*, 2015, **137**, 6440–6443) is really more appropriate to cite.

Response: We have added the Knowles JACS 2015 references as requested. We have chosen to keep the Studer reference due to the connection to p-block radical chemistry.

I am not sure that I agree with the description of the basis for coordination induced bond weakening, especially the use of invariably. While in most instances coordination of a small molecule to a low-valent metal metal reduces the pKa of the bound of the bound ligand, in some cases it is quite modest. In other cases, significant X-H bond-weakening occurs at less proximate sites although the impact of increasing acidity is almost non-existent (ie coordination of amides to Ti(III), Sm(II), etc.)

Response: We have replaced “invariably” with “often”. After that change, with all due respect to Reviewer 2, we feel that our description is accurate as written.

Sm-water bond-weakening was established by Flowers several years earlier (2015) than the cited Mayer report (*J. Am. Chem. Soc.* **2015**, *137*, 11526-11531). The estimate of bond-weakening by Flowers described in a follow-up small review (*Dalton Trans.* **2019**, *48*, 16142-16147) is smaller but consistent with the value determined by Mayer and Kolmar employing thermochemical cycles but is more relevant to synthetic systems in organic media. The value by Mayer is an overestimate since aqueous potentials and pKa's were used not relevant to the solvent employed in the reduction of an enamine in THF. I believe this was also recently pointed out by Peters and coworkers in a recent *JACS* publication as well. This should be corrected in the introduction and in the results and discussion section below Figure 3. It

isn't necessary to get into this level of detail, but certainly the range of bond weakening based on *experimental* evidence can be described briefly.

Response: Thank you to Reviewer 2 for the helpful suggestions. We have updated this part to include a range of BDFE values (26-39 kcal/mol) for aqueous Sm(II) and added the requested citation of Flowers.

In Figure 1 a, the incorrect structure is shown. Upon addition of water to low-valent titanocene, chloride ions are solvated by water and displaced to the outer sphere. A better representation

is shown above and supported by EPR, voltammetry, and computational studies carried out by Gansauer and coworkers (*Angew. Chem. Int. Ed.*, 2012, **51**, 3266–3270). I recommend this structure be shown.

Response: Thank you. The figure has been updated, and the ACIE 2012 by Gansauer et al. is now cited.

The discussion around Figure 1b should be referred to as multisite PCET. This terminology is well-established for the description of biological systems such as the OEC of photosystem II and synthetic systems. 1b is also a classic example of acidification of a bound small molecule coupled to ET from a redox center in the language used to describe coordination induced bond weakening.

Response: Thank you to Reviewer 2 for correcting our terminology. The “multisite PCET” term has been added where appropriate, which also helps address a concern of Reviewer 1.

In the second paragraph of the results and discussion, it may be useful to cite recent work of Knowles on the coordination induced bond weakening of cyclopropanes (*J. Am. Chem. Soc.* 2022, 144, 34, 15488–15496) since in some ways it is analogy to the current system.

Response: Thank you. This work is now cited in the third paragraph of Results & Discussion.

In the weakening of N-H bonds, recent work has demonstrated significant weakening in Sm(II)- NH₃ N-H bonds (*J. Org. Chem.* **2022**, 87, 1689-1697).

Response: Thank you, this work is now cited in the Introduction.

In the conclusion, the authors discuss the extension to the weakening of C-O bonds. I think this should be reworded. In the reduction of epoxides by low-valent titanocene, coordination induced bond weakening is an important feature of the first step in reduction. Although it doesn't involve X-H bond weakening, coordination of an epoxide to Ti(III) clearly weakens the C- O bond and is a classic example of coordination induced bond weakening. There needs to be more context in the description of the present Al-Fe system to differentiate from more classic systems.

Response: We added a clause stating that our system is “a metal/ligand cooperative analogue of established Ti^{III} chemistry” and cited a recent review on $Ti(III)$ radical relay catalysis.

Reviewer #3 (Remarks to the Author):

Coordination-induced O–H/N–H bond weakening by a redox non-innocent, aluminium-containing radical.

Mankad and coworkers report a remarkable discovery based on coordination-induced bond weakening of protic substrates on coordination to a aluminium radical. This radical is derived from the homolysis of a Fe–Al heterometallic. The authors have provided convincing evidence not only for radical generation (which expands and builds upon prior work from the group, e.g. *J. Am. Chem. Soc.* 2022, 3210) but also that a PCET event is at play, rather than simply an acidification of O–H or N–H bond on coordination.

The results are important and will find broad interest as they suggest a general strategy for use of H₂O, NH₃, simple alcohols and amines through activation at main-group radical intermediates. Ultimately the information could inform design of new catalysts. The authors are also correct in stressing then key point of novelty that this is the first time such reactivity has been observed at aluminium, the most abundant metal in the earth’s crust.

I am very supportive of publication and have the following suggestions.

Response: Thank you to Reviewer 3 for the supportive evaluation and enthusiastic comments.

(1) The narrative of the current draft is a little awkward. The work is framed in terms of PCET but the bulk of the initial results and discussion, captured in Figure 2, focuses on reactivity of heterometallics. At times this feels like two stories. Can the authors edit this section to reduce the amount of content, or better draw the connection to the PCET results – which are the key novelty.

Response: We feel that the data in Figure 2 is critical to the story, as without it there is a less than convincing case that X-H activation involves radical intermediates. To help make the connection clear for the reader, we have made some minor edits to this section.

(2) Additional analysis of the electronic structure of A by DFT, including a spin-density plot would help argue the ligand-based character of the radical.

Response: Thank you, this has now been added to revised Figure 4.

(3) The estimation of the BDE of [A–H₂O] radical is a little unsatisfying. I think it is appropriate to put an upper-bound on this, but the lack of correlation between the computational values and experimental data in benchmarking of model is a bit worrisome. The use of diffuse functions in the basis-set is known to be important to properly model metal hydride complexes, I would strongly suggest that the authors investigate Ahlrich’s basis set that includes both polarisation and diffuse e.g. def2-QZVPPD or def2-TZVPD and compare data to those in Table S1. The functional group dependence should also be explored including both hybrid and mGGA functionals.

Response: We thank the reviewer for suggesting we consider the methodology of the bond energy estimate. We have deleted eq 1-3 and so this discussion has been removed. Instead, we directly evaluated the bond dissociation enthalpy and BDFE of the Al-water O-H bond. We did this with M06 and PBE1PBE functionals with the def2-TZVPD basis set. The energies for this bond converged going

from TZVP to TZVPD. Additionally, for a model ligand complex (removal of the large aryl groups that play a very minor role in the chemistry) we calculated the CCSD(T)/def2-TZVP value and this method suggests that DFT overestimates the O-H bond energy, which provides confidence in reporting the DFT values.

(4) It is not uncommon to not be able to observe ^{13}C resonance of sites directly bonded to Al by 1D NMR, but usually these are resolved by 2D methods. Can the authors find these for 6a-e by ^1H - ^{13}C HSQC experiments.

Response: Thank you for the suggestion. We were, indeed, able to observe the ^{13}C NMR resonance of the methyl directly attached to aluminum using ^1H - ^{13}C HMQC 2D NMR spectroscopy. We have added HMQC data for 6b-e in the Supporting Information. Because 6a is already reported in literature, we did not add HMQC data for it.

(5) Can ^{183}W satellites not be observed in the NMR data of 4? If not a note should be added to the SI to clarify.

Response: For complex 4, yes, we do observe ^{183}W satellite peaks. In the revised Supporting Information, we have added a magnified view of the carbonyl region to show the satellite peaks distinctively.

(6) There is a serious issue with the characterisation data reported for 6d. This compound does not contain W, something is wrong here. Please check the data and assignments.

Response: Thanks for pointing this out. We have added new spectra without tungsten impurities for 6d.

Reviewer #4 (Remarks to the Author):

The structural data presented by the authors does support their conclusion, but RS15 and SS1 need refining with sensible cutoff on the data resolution, see comments below. For compound 2 the description in the manuscript of the numbering and bond distances are not consistent with the CIF, I think the atom numbers were changed at somepoint, see below. Other standard information is missing from the CIFs that the authors need to fill in.

RS15

While the data was collected to 0.55 angstroms, data is considered observed is $I/\sigma(I) > 2$, see INTENSITY STATISTICS FOR DATASET below. Therefore from ~ 0.72 to 0.55 angstroms the data is on average unobserved. This mean a massive amount of noise was added to this data set. If the data are cut at 0.72 Angstroms then the high electron density peaks around the W drop from ~ 4 electrons to ~ 1.5 electrons. In cutting the data all the AlertA and B's are sorted out and do not occur and leading to a better refined structure.

Why are there no hydrogen atoms on the Cyclopentadienyl ligands? Why are its hydrogen not included in the chemical formula?

Please fill out “_exptl_crystal_colour ?” in the CIF even if the crystal is colourless.

Response: Thank you for the helpful suggestions. All these changes were implemented, which did indeed clear all level A and B errors. Revised CIF and checkCIF are enclosed.

RS06_m

b cell parameter is missing a decimal place, give as 18.481 but error is given as 0.0003 why?
Please add the colour and size of the crystal into the CIF.
Please label the atoms consistently with the other structures and not just use the number straight out of SHELXT.

Response: Thank you, all these changes were made.

SS1

While the data was collected to 0.50 angstroms, data is considered observed is $I/\sigma(I) > 2$, see INTENSITY STATISTICS FOR DATASET below. Therefore from ~0.71 to 0.50 angstroms the data is on average unobserved. This mean a massive amount of noise was added to this data set. Also R(merge) even at 0.71 angstroms is 63.96% which again is adding more noise to the data. From my experience R(merge)s of greater than 30% stop adding more noise than useful data. The authors may want to go back at look at the processing of the original data and determine if the crystal was dying in the beam and if possible cut the data at that point. However, if the data were cut at 0.82 Angstrom all the AlertA and B's are no longer triggered.

Was the temperature of the data collection really 273K?

Please add crystal dimension into the CIF.

Please add the following into the CIF

_diffn_source 'Advanced Photon Source, NSFs ChemMatCARS'

Response: Thank you, all of these changes were made, and level A and level B errors are indeed gone.

Please number this atom consistently with those of the other structures C00Q

Please check:

“The X-ray crystal structure of 2 indicates localized π -bonding consistent with disruption of pyridine aromaticity: the C2-C3 and C5-C6 distances show double-bond character [1.341(2)-1.343(2) Å] while the C3-C4 and C4-C5 distances show single-bond character [1.510(2)-1.516(3) Å], and the N-C2 and N-C6 distances are elongated [1.396(2)-1.400(2) Å] compared to pyridine (1.340 Å)²⁶. The C4-C4' distance is also indicative of single bonding [1.569(3) Å], consistent with the pyramidalized, C(sp³)-like geometries at these centers.”

The bonds listed above are not consistent with the number in the CIF provided, as I believe these should be in the thirties, C2-C3 are in the backbone of the main ligand.

Response: With due respect to the reviewer, our intention here was to refer to the numbering system that is standard in the organic chemistry field for nitrogen heterocycles, i.e., the C2 position of a pyridine, etc. However, we did update the bond distances to account for the new refinement.

Reviewers' Comments:

Reviewer #1:

Remarks to the Author:

All my concerns have been satisfactorily addressed.

Reviewer #2:

Remarks to the Author:

I have carefully read the reviewer comments and revised manuscript. In my view, Professor Mankad and coworkers have addressed reviewer comments and concerns. I am pleased to recommend acceptance in the strongest terms.

Reviewer #3:

Remarks to the Author:

The authors have carefully considered and addressed the reviewer comments. The new and expanded computational study greatly improves the manuscript. The work should be published without further delays.

Reviewer #4:

Remarks to the Author:

The changes have been made as I requested to the crystallographic refinements and have produced better models with less issues. These models still support the authors discussions and conclusions.

On a minor point could you please include crystal dimensions, in the CIF, for RS06 and RS15.